# Cross-Cultural Adaptation and Validation of the “Brief Scale of Perceived Barriers to Physical Activity for Children”: Analysis of Psychometric Properties

**DOI:** 10.3390/healthcare13222991

**Published:** 2025-11-20

**Authors:** Raquel Pastor-Cisneros, María Mendoza-Muñoz, Amparo Rodríguez-Gutiérrez, Jorge Carlos-Vivas

**Affiliations:** 1Physical Activity for Education, Performance and Health (PAEPH) Research Group, Faculty of Sport Sciences, University of Extremadura, 10003 Cáceres, Spain; raquelpc@unex.es (R.P.-C.); jorgecv@unex.es (J.C.-V.); 2Department of Communication and Education, Universidad Loyola Andalucía, 41704 Sevilla, Spain; 3Research Group on Didactic and Behavioural Analysis in Sport, Faculty of Sports Sciences, University of Extremadura, 10003 Cáceres, Spain; arodriguezgu@unex.es

**Keywords:** cultural adaption, exercise, motivation, physical literacy, psychometric properties, SPLA-C

## Abstract

**Background**: Physical activity (PA) provides significant health benefits, yet inactivity remains high in Spain, especially among adolescents and increasingly in children. Identifying barriers to PA is essential, but available tools are mainly designed for adolescents. This study aimed to adapt the “Brief Scale of Perceived Barriers to Physical Activity” for Spanish schoolchildren aged 6–12 and examine its validity and reliability. **Methods**: The “Brief Scale of Perceived Barriers to Physical Activity for Children” was linguistically and culturally adapted. Comprehension was assessed through cognitive interviews, and reliability was examined via a test–retest procedure with 137 Spanish schoolchildren. Several analyses were conducted, including confirmatory factor analysis (CFA) to assess the factor structure, along with reliability metrics: Cronbach’s alpha (α) for internal consistency and the intraclass correlation coefficient (ICC) for test–retest reliability. **Results**: CFA confirmed a four-factor structure (self-concept, motivation–interest, social support, and task incompatibility) in a sample of 137 with excellent fit indices (χ^2^/df = 1.394, RMSEA = 0.054, CFI = 0.976, TLI = 0.966). Internal consistency ranged from good to excellent (α = 0.831–0.979). Temporal stability was substantial to near perfect (ICC = 0.708–0.979). Measurement error was low for all items and the total score (SEM% = 6.1–37.2; MDC% = 17.0–103.0), demonstrating accuracy. **Conclusions**: The “Brief Scale of Perceived Barriers to Physical Activity for Children” was proven to be a reliable and valid tool for assessing perceived barriers to PA in Spanish children. It offers developmentally appropriate insights that can guide strategies to enhance supportive environments and promote long-term active behaviours. As part of the social domain, it contributes to the Spanish Physical Literacy Assessment for Children (SPLA-C) model, the first physical literacy (PL) assessment instrument developed in Spain.

## 1. Introduction

Physical activity (PA) is widely recognised as one of the key determinants of physical, psychological, and social well-being across the lifespan [1]. Regular participation in PA contributes to the prevention of chronic diseases such as obesity, cardiovascular disease, type 2 diabetes, and certain types of cancer, as well as improved musculoskeletal health, cognitive functioning, and emotional regulation [2,3,4]. Moreover, engaging in PA from an early age promotes healthy growth and development, better academic performance, and long-term adherence to active lifestyles [5,6].

Despite this extensive evidence, data on PA levels in the Spanish population are alarming: approximately one-third of individuals do not meet the healthy PA levels recommended by the World Health Organisation (WHO) [7]. This situation is particularly concerning among young people, given that physical inactivity during the early years can persist into adulthood [8]. While childhood is generally characterised by higher PA levels compared to adolescence, recent decades have shown a worrying decline even among children [9]. During adolescence, this decline becomes more pronounced, marking a critical stage for the establishment of sedentary habits [10]. In this context of widespread inactivity, researchers have sought to identify the main barriers that prevent or hinder PA participation [11,12]. Evidence has emerged that perceived barriers vary according to the population to which they are administered [13,14]. These perceived barriers in childhood can vary depending on factors such as age, gender, and cultural context [15]. Therefore, developing and validating instruments that are adapted to the characteristics of specific populations is essential to accurately assess such barriers.

A number of questionnaires and scales have been developed to assess barriers to PA in different populations [16,17]. To date, no validated Spanish-language instrument exists to assess perceived barriers to PA in children. The “Brief Scale of Perceived Barriers to Physical Activity for Children” [18], originally designed for adolescents, was selected for its concise structure and prior use in Spanish-speaking contexts. A cultural and developmental adaptation was therefore conducted to ensure suitability for younger populations. Although other instruments, such as the questionnaire developed by Arlinghaus et al. [19], assess facilitators and barriers in children, this tool has not been validated in Spanish and includes a greater number of items and subscales. The “Brief Scale of the Perception of Barriers to the Practice of Physical Activity” [18] is a 12-item self-report instrument designed to identify the barriers that adolescents face when participating in organised sports. Adolescents are asked to indicate on a Likert scale (1: “strongly disagree” to 5: “strongly agree”) the extent to which they perceive the different items as barriers to participating in organised sports activities. However, as the target population of this study was children aged 6 to 12, covering grades 2 to 6 of primary school, there is an absence of studies that have included an adaptation for these ages, thus leaving a gap in the existing literature. In this regard, it is necessary to adapt the original instrument rather than use it directly, given the existence of socio-cultural and contextual differences [20,21] as well as cognitive and linguistic differences [22] between the adolescent population and school-age children. This approach will facilitate a more precise evaluation of the perceived barriers to PA among schoolchildren. Furthermore, given the decline in PA levels during adolescence [10], a stage in which the barriers to PA have been studied using this scale [18], there is a need to adapt this same instrument to the primary school stage. This would make it possible to study the barriers to PA that exist prior to adolescence and thus be able to establish a comparison between both populations.

In light of the substantial disparities in development that are observed between school-aged children and older populations [22], it is imperative to devise a version of the “Brief Scale of Perceived Barriers to Physical Activity” that is more accessible and comprehensible to children, with adjustments being made to the language, structure, and items. The aim of this study is twofold: firstly, to adapt the “Brief Scale of Perceived Barriers to Physical Activity” instrument for its use with Spanish schoolchildren aged 6 to 12 years, and secondly, to evaluate the validity and reliability of the adapted instrument. It is hypothesised that the adapted version will demonstrate adequate psychometric properties, showing satisfactory validity and reliability for assessing perceived barriers to physical activity in this population.

## 2. Materials and Methods

### 2.1. Study Design

The present study constitutes an instrumental and methodological investigation into the linguistic and cultural adaptation of the “Brief Scale of Perceived Barriers to Physical Activity” for use with children aged between 6 and 12 years. The original questionnaire was developed for adolescents (aged 12–18); an adapted version was created for children (Brief Scale of Perceived Barriers to Physical Activity for Children). The 6–12 age range was selected to cover the entire primary education stage. Moreover, the study examined the psychometric properties (validity and reliability) of the adapted questionnaire in this population.

The study was conducted in three phases.
(1)Linguistic and cultural adaptation: The vocabulary, syntax, and content of the original instrument were modified to align with the cognitive, linguistic, and socio-emotional developmental levels of school-aged children. Each item underwent an adaptation or reformulation process, followed by a review by three experts in the fields of children’s physical literacy (PL), education, and motivation in PA.(2)Assessment of comprehension through cognitive interviews: To evaluate the clarity and functional understanding of the adapted items, individual cognitive interviews were conducted with a pilot sample of schoolchildren (n = 17). The students were recruited through the physical education teachers of each year group, and the interviews were conducted in person in the classroom. Initially, 20 children from 2nd to 6th grade (four from each year) were contacted, but three children did not attend class on the day of the interviews. Thus, the final sample consisted of 17 schoolchildren: four pupils in Year 2, four pupils in Year 3, and three pupils in each of the remaining years. These interviews focused on analysing both literal and contextual comprehension. Understanding was also evaluated through an interview process aligned with best practices to ensure child-friendly instruments [23].(3)Psychometric analysis and test–retest reliability: In the second phase, the psychometric properties of the “Brief Scale of Perceived Barriers to Physical Activity for Children” were analysed. This included a confirmatory factor analysis (CFA) to assess construct validity. Additionally, a test–retest reliability study was carried out with a sample of children (n = 137) who completed the questionnaire twice, with a two-week interval, to assess the temporal stability of the responses. This rigorous, phased approach ensured that the “Brief Scale of Perceived Barriers to Physical Activity for Children” was not only linguistically and culturally appropriate for Spanish children but also psychometrically sound in terms of content validity, functional comprehension, and reliability.(4)This structured process ensured that the instrument was rigorously adapted to children’s developmental needs, enabling the assessment of content validity, functional comprehension, and temporal consistency. The final version of the adapted instrument is available in Appendix A.

### 2.2. Participants

A total of 137 schoolchildren (68 boys and 69 girls), aged 6 to 12 years, were recruited from several primary schools in Extremadura, Spain, using non-probability convenience sampling. Participants were required to meet the following inclusion criteria: (1) be enrolled in primary education; (2) provide informed consent signed by a parent or legal guardian; and (3) demonstrate sufficient reading comprehension to complete the adapted questionnaire.

The 6–12 age range was chosen to represent the full span of primary education in Spain, aligning with the intended application of the “Brief Scale of Perceived Barriers to Physical Activity for Children” within the Spanish Physical Literacy Assessment for Children (SPLA-C) model. The instrument was designed to be understandable and relevant for all children in this stage, regardless of grade or developmental level. Cognitive interviews confirmed that the items were comprehensible across the entire age spectrum. First-grade students were excluded, as those assessed early in the academic year had not yet completed the physical education (PE) curriculum, which could limit their understanding of sport-related contexts. The overarching goal was to develop a unified tool applicable to the whole primary school population with prior PE exposure, ensuring consistent use in both research and educational practice.

### 2.3. Ethics Approval

This study involved human participants and was approved by the Bioethics and Biosafety Committee of the University of Extremadura (approval number 288/2024). The updates of the Declaration of Helsinki, amended by the 75th General Assembly of the World Medical Association (Helsinki, Finlandia, 2024) and Law 14/2007 on Biomedical Research, were followed. The participants provided informed consent to participate in this study before taking part.

### 2.4. Instrument and Adaptation Process

The data were collected via an adapted version of the original “Brief Scale of Perceived Barriers to Physical Activity for Children” [18], which is a 12-item self-report questionnaire developed to identify barriers that adolescents encounter when engaging in organised sports. The questionnaire comprises 12 items distributed across four subscales: self-concept, motivation interest, social support. and incompatibility tasks. Participants rate each item on a Likert scale ranging from 1 (“*strongly disagree”*) to 5 (“*strongly agree”*), indicating the extent to which they perceive each factor as an obstacle to participation in organised sports activities. To adapt the instrument for use with children, the items in the “Brief Scale of Perceived Barriers to Physical Activity for Children” underwent a process of linguistic and cultural adaptation (see Table 1). These adaptations consisted of: (a) simplification of vocabulary and syntax; and (b) elimination or replacement of abstract or difficult-to-interpret concepts. The adaptation of the “Brief Scale of Perceived Barriers to Physical Activity” resulted in the “Brief Scale of Perceived Barriers to Physical Activity for Children”, which retained the original theoretical dimensions. Since the adaptation was based on an established structure, a CFA was conducted to verify the validity of the original structure in the new population. A panel of experts in the field of PL and PE in primary schools reviewed the content of each item, and cognitive interviews were conducted with a pilot group of 17 children to verify the clarity, comprehension, and appropriateness of each item.

During the cognitive interview, it was determined that the participants experienced difficulties in comprehending the instructions prior to the initial sentence. Consequently, these were removed to improve clarity. Regarding the items, the scale was rated as clear and understandable, except for items 2, 3, 4, 6, and 7, which underwent slight modifications in vocabulary or syntax (see Table 1). These adjustments were minor and did not alter the conceptual meaning of the items or the factorial structure of the original instrument.

### 2.5. Statistical Analyses

All the data gathered were recorded in a database designed for this research project, with personal data kept anonymous. Statistical analyses were performed using the Statistical Package for the Social Sciences (SPSS, version 25.0; IBM SPSS Inc., Armonk, NY, USA). CFA was performed using the software package AMOS v.23.0.0 (IBM Corporation, Wexford, PA, USA). The various items of the “Brief Scale of Perceived Barriers to Physical Activity for Children” were incorporated as elements. In order to assess the model’s goodness of fit, the following indices were selected: (1) the chi-square probability with appropriate non-significant values (*p* > 0.05) [24], (2) the root mean square error of approximation (RMSEA) [25], (3) the comparative fit index (CFI), (4) the Tuker–Lewis index (TLI), and (5) the chi-square per degree of freedom ratio (CMIN/DF) [26].

Subsequent to this, a test–retest reliability check was performed a period of fifteen days later. The data are presented as the mean and standard deviation (SD) for the initial and follow-up assessments. The Shapiro–Wilk and Levene tests were employed to verify the normality and equality of variance for all the assessed variables. The internal consistency and reliability of each item in the SMOSS and the overall score were evaluated using Cronbach’s alpha coefficient. According to Glen (2022) [27], Cronbach’s alpha can be interpreted as follows: 0.5 signifies unacceptable; 0.5–0.6 indicates poor; 0.6–0.7 suggests questionable; 0.7–0.8 is acceptable; 0.8–0.9 is good; and 0.9 represents excellent. The reliability or consistency of the test was evaluated by computing the intraclass correlation coefficient (ICC) with a 95% confidence interval [28]. A two-way random effects model, single measures, absolute agreement, and ICC were employed to demonstrate the degree of agreement between the two tests. The interpretation of ICC scores was conducted in accordance with the standards established by Landis and Koch (1977) [29]: <0.20 indicates slight agreement, 0.21–0.40 fair, 0.41–0.60 moderate, 0.61–0.80 substantial, and 0.80 near perfection.

Furthermore, the standard error of measurement (SEM) and the minimum detectable change (MDC) [30] were utilised to evaluate absolute reliability. Finally, Spearman’s correlation was performed to analyse the relationship between each item and the total score. The significance level was set at *p* ≤ 0.05 for all tests.

## 3. Results

### 3.1. Confirmatory Factor Analyses (CFA)

A CFA was conducted with a total of 137 participants aged 9.40 (±1.86) years, of whom 48.2% were boys and 51.8% were girls. The resulting model from the CFA is displayed in Figure 1. To improve model fit, two theoretically justified residual covariances were included within the *Self-concept* factor: between Item 8 (*“Soy peor que los/las demás en el deporte”*) and Item 6 (“*Porque me da vergüenza mi cuerpo cuando practico deporte”*), and between Item 11 (“*Porque tengo miedo a que se rían de mí*”) and Item 4 (“*Porque no estoy en buena forma física*”). These covariances were allowed because the items share specific wording and conceptual overlap related to negative self-perceptions of physical competence and body image. In both cases, the residual correlations reflect shared variance beyond the general self-concept factor, which is theoretically expected in measures of self-conscious emotions and perceived competence in sport contexts.

The CFA indicated an excellent model fit. The chi-square divided by degrees of freedom (CMIN/df) fell well below the recommended threshold of 2.0, suggesting a good balance between model complexity and data representation [31]. The chi-square test was marginally significant, a result that is not uncommon in moderately sized samples and does not necessarily imply poor fit [32]. The CFI and the TLI exceeded the recommended threshold of 0.95 for excellent fit [33]. The RMSEA, indicating a low level of approximation error and falling well within the acceptable range, was below 0.06 [34]. These results provide strong empirical support for the adequacy of the proposed factorial structure (see Table 2).

### 3.2. Test–Retest Reliability and Internal Consistency

Table 3 displays the internal consistency, reproducibility, and systematic differences of the “Brief Scale of Perceived Barriers to Physical Activity for Children”. Overall, the internal consistency ranged from good to excellent for all items and total score of the scale (Cronbach’s α if item deleted from 0.831 to 0.979). All items at the initial and follow-up tests significantly correlated with total score (rho: 0.400 to 0.608).

Reproducibility outcomes revealed substantial to near perfection test–retest reliability for each item and the total score of “Brief Scale of Perceived Barriers to Physical Activity for Children” (ICC from 0.708 to 0.979). The SEM and SEM% values for each item and the total score ranged from 0.20 to 1.02 and from 6.1 to 37.2, respectively. Likewise, the MDC and MDC% values for each item and the total score ranged from 0.56 to 2.83 and from 17.0 to 103.0, respectively.

Finally, comparison outcomes overall showed no significant differences for any of the items (*p* > 0.05), except for item 2 (*p* = 0.024).

## 4. Discussion

The objective of this study was to adapt the “Brief Perceived Barriers to Physical Activity Scale”, which was originally designed for adolescents, for use with Spanish schoolchildren aged 6–12. The study also examined the scale’s validity and reliability. The study also analysed the psychometric properties of the adapted instrument, with a focus on its factor structure, internal consistency, and temporal stability. The findings of the present study lend support to the validity and reliability of the adapted instrument for use with this age group. The CFA demonstrated an optimal alignment between the model and the data, thereby suggesting that the theoretical structure of the questionnaire is well-suited to this age group. Reliability analyses indicate that the instrument demonstrated satisfactory internal consistency for all items, as well as high temporal stability for the total score. The findings of this study indicate that the proposed adaptation is a valuable instrument that will enhance the comprehension and evaluation of perceived barriers to PA among schoolchildren, thereby providing a robust foundation for its utilisation by teachers and researchers.

The CFA analysis of the “Brief Scale of Perceived Barriers to Physical Activity for Children” revealed a perfect fit between the theoretical model and the empirical data, thereby supporting the structural validity of the scale in schoolchildren. Specifically, the CMIN/DF value (1.394) fell within the range indicating an excellent fit (<2), the CFI (0.976) and TLI (0.966) indices exceeded the threshold of 0.90, and Hu & Bentler’s [33] criterion of excellence (0.95) was met. Additionally, the RMSEA value (0.054) remained well within the acceptable range (<0.08), indicating a reasonable approximation error. The chi-square test was marginally significant (*p* = 0.040), which frequently occurs in medium-sized or large samples and does not necessarily constitute sufficient evidence of a poor fit [32]. Consequently, it can be concluded that the indices obtained from the original factorial structure of the instrument remained adequate after its linguistic and conceptual adaptation for use with children aged 6 to 12 years. The robustness of the obtained indices can be partly explained by the strategy of simplifying the wording of the items, which facilitates adequate comprehension by children in instruments focused on children’s perception [35]. Therefore, the appropriateness of the factorial model suggests that schoolchildren could comprehend and respond uniformly to the various items that constitute the adapted scale. This lends further credence to the questionnaire as a valid instrument in primary education, thereby facilitating the acquisition of data on salient issues in PA, including the identification of perceived barriers from the perspective of the child [36,37].

In terms of the internal consistency of the “Brief Scale of Perceived Barriers to Physical Activity for Children”, the total score showed excellent consistency (Cronbach’s α = 0.979). The internal consistency of all items and the total score of the scale varied in terms of excellence (Cronbach’s α if item deleted: 0.971–0.985). The results indicate a high degree of homogeneity among the items and their ability to reliably measure the construct of perceived barriers to engaging in PA. Furthermore, the subscales that constitute the adapted instrument have previously been assessed alongside the evaluation of children’s perceptual and emotional constructs and PA [38,39]. Overall, most items exhibited alpha values that approached the optimal level of excellence, thereby reinforcing the instrument’s internal consistency. In general, the majority of items demonstrated excellent values (Cronbach’s α if item deleted >0.900), with items 2, 4 and 7 exhibiting slightly lower values (Cronbach’s α if item deleted =0.866; 0.884; and 0.831, respectively), which remained high. The scale’s internal consistency was very strong, with values higher than those reported in a previous study that also measured perceived barriers to PA in a different population (aged 18–35) [40].

The evaluation of the test–retest reliability of the “Brief Scale of Perceived Barriers to Physical Activity for Children” revealed positive results in terms of the temporal stability of the instrument. The total score of the questionnaire yielded an ICC of 0.979, indicating an excellent degree of agreement between measurements taken fifteen days apart. This finding lends further support to the robustness of the instrument for reliably assessing perceived barriers to PA, and presented higher ICC values than other studies related to barriers to PA [41,42].

Further corroboration of the instrument’s overall accuracy was provided by absolute error analyses (SEM and MDC), with minimal measurement error in the total score (SEM% = 6.1; MDC% = 17.0). However, two individual items presented values closer to the unstable SEM% and MDC% thresholds: item 7 with an SEM% of 37.2, and item 2 with an SEM% of 33.4%. This could reflect greater variability in the children’s responses [43]. Nevertheless, the obtained values demonstrate a high degree of suitability for schoolchildren, as evidenced by the consistent responses observed.

The investigation into the interrelationships between each component and the aggregate score of the adapted scale yielded favourable outcomes, consistent with those of a previous reliability and validation assessment of the instrument in adolescents with type 1 diabetes [44]. All components exhibited moderate to strong significant correlations with the total questionnaire score (*p* < 0.01), thereby substantiating the notion that each component substantially contributes to assessing the overall construct. The Spearman’s rho coefficients ranged from 0.400 to 0.608, indicating a moderate to strong relationship between individual items and the overall score. These results highlight the instrument’s internal consistency and its ability to evaluate the construct of interest.

### 4.1. Strengths and Practical Implications

A notable strength of the adaptation of the “Brief Scale of Perceived Barriers to Physical Activity for Children” to the Spanish context is the rigorous linguistic and cultural adaptation process that was followed. The scale was meticulously calibrated to ensure that its language, structure, and contents were comprehensible and pertinent to children between the ages of 6 and 12, with consideration for their cognitive, linguistic, and socio-emotional characteristics.

Another salient strength is its psychometric soundness. CFA demonstrated an excellent fit of the model, and internal consistency reached good to excellent values, along with substantial to near-perfect test–retest reliability. These findings provide support for the stability and accuracy of the questionnaire in measuring perceived barriers to PA in children.

In practical terms, the existence of a brief, clear and reliable tool facilitates its use in school and community settings. In the context of primary education, a pivotal stage in the establishment of healthy habits, this scale can be utilised by PE teachers, counsellors, coaches or health professionals with a view to the following:(1)It is imperative to ascertain the individual and contextual barriers that impede PA among schoolchildren. Such barriers may include a lack of time, a perceived difficulty, or environmental influences.(2)It is imperative that educational interventions and PA promotion programmes are designed with greater consideration for the specific needs of children.(3)The impact of these interventions should be assessed over time, both within educational institutions and within community or extracurricular programmes.

The brevity and accessible format of the questionnaire reduce the risk of fatigue or disinterest during completion, which is an additional practical advantage for its application with children.

Importantly, this study aligns with a recent article on the development of the first assessment model for physical literacy in Spain: the Spanish Physical Literacy Assessment for Children (SPLA-C) [45]. In this Delphi study, national experts concluded that this scale should form part of the ‘social interaction and barriers to physical activity practice’ component of the new PL assessment model in Spain. As part of the SPLA-C model, the “Brief Scale of Perceived Barriers to Physical Activity for Children” enables schools and policymakers to monitor barriers to PA among schoolchildren in relation to the PL. This information can be used to design more holistic PE programmes that not only address physical skills, but also accessibility and adaptation, enabling schoolchildren to participate in sports.

### 4.2. Limitations and Future Line Research

Whilst the findings lend support to the validity and reliability of the “Brief Scale of Perceived Barriers to Physical Activity for Children” for use with children, it is important to acknowledge the following limitations, which should be considered when interpreting the results and guiding future research.

Firstly, the potential for variations in the instrument’s structure or reliability based on gender, age, or sporting experience was not investigated. It is recommended that subsequent research endeavours seek to augment the sample size and its geographical diversity, with the objective of substantiating the validity of the “Brief Scale of Perceived Barriers to Physical Activity for Children” across a more extensive and heterogeneous population. Furthermore, conducting exploratory and CFA within specific subgroups (e.g., by age, gender, or level of sports participation) would assist in identifying potential structural variations.

Secondly, the utilisation of a self-report format engenders inherent challenges when conducting research with young children, as their capacity for self-reflection and interpretation of abstract statements can vary considerably. Despite the implementation of cognitive interviews during the pilot phase, comprehension difficulties were still observed for certain items (2, 3, 4, 6, and 7). This finding indicates that while the overall linguistic adaptation was deemed satisfactory, there is a need for further refinement in certain areas. It is recommended that future studies incorporate more detailed qualitative approaches, such as interviews or focus groups with children, with a view to exploring and improving the clarity of problematic items. Furthermore, the creation of illustrated or visually supported versions could enhance accessibility, particularly for younger children or those with limited reading comprehension skills.

This study did not assess criterion validity, which limits the external validity and generalisability of the findings to other populations or settings. Moreover, convenience sampling was used in this study, which is suitable for validity assessment but may limit the generalisability of the results to the broader population of Spanish schoolchildren.

It is evident that the cross-sectional design of the study constitutes a limitation, underscoring the necessity for subsequent longitudinal research to investigate the responsiveness of the “Brief Scale of Perceived Barriers to Physical Activity for Children” to educational interventions or motivational programmes. Furthermore, it is imperative to ascertain the extent to which the “Brief Scale of Perceived Barriers to Physical Activity for Children” can discern significant alterations in in perceived barriers to PA practice over time.

## 5. Conclusions

The “Brief Scale of Perceived Barriers to Physical Activity for Children” demonstrated a valid factor structure, good to excellent internal consistency, and substantial to near-perfect temporal reliability. These psychometric properties support its use as a reliable tool for assessing perceived barriers to physical activity in school-aged children. By facilitating understanding of the obstacles children face in engaging with PA, the scale can inform the design of effective strategies to reduce these barriers in educational and community settings. It represents a valuable resource for research and practice in paediatric exercise science, supporting the promotion of inclusive and health-oriented practices from an early age. The “Brief Scale of Perceived Barriers to Physical Activity for Children” is part of the social domain of the SPLA-C, the first model for the assessment of PL in Spain.

## Figures and Tables

**Figure 1 healthcare-13-02991-f001:**
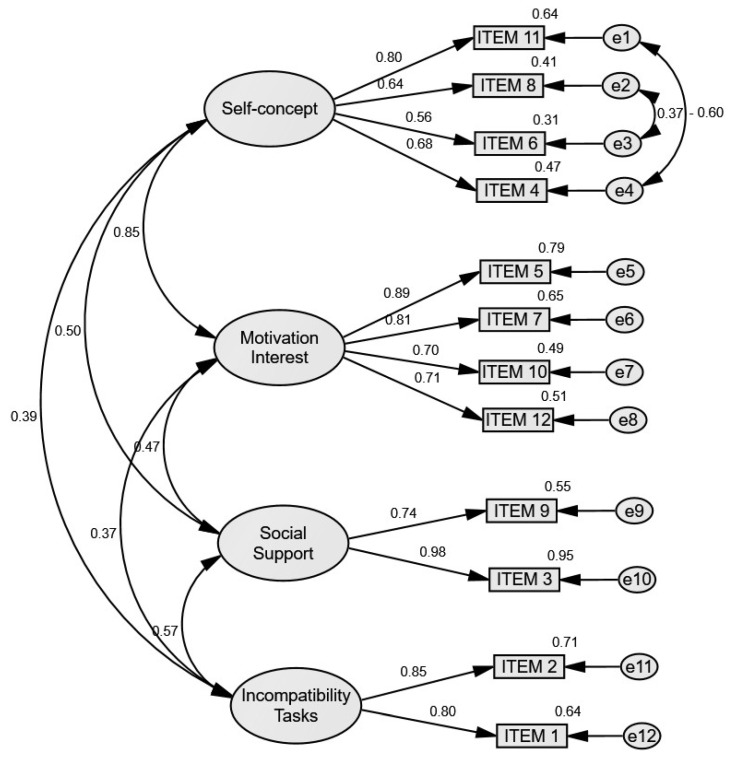
CFA resulting model for “Brief Scale of Perceived Barriers to Physical Activity for Children”.

**Table 1 healthcare-13-02991-t001:** Different versions of the “Brief Scale of Perceived Barriers to Physical Activity for Children”.

Items	Original Version	Version 1. Agreed Version Through Internal Consensus	Final Version. Adaptations After Cognitive Interviews
Instructions	*Por favor, valora en qué medida los siguientes motivos son causas o barreras para no participar en alguna actividad física dirigida fuera del colegio o instituto (como puede ser en clubes, ayuntamientos, centro cívico, gimnasio,* etc.)	*Por favor, valora en qué medida los siguientes motivos son dificultades que te impiden practicar deporte/actividad física fuera del colegio o instituto (como puede ser en clubes, ayuntamientos, centro cívico, gimnasio,* etc.)	-
Opening sentence	*No participo en alguna actividad física fuera del colegio o instituto…*	*No participo en alguna actividad física o deporte fuera del colegio porque…*	*No participo en alguna actividad física o deporte fuera del colegio porque…*
1	*Porque tengo muchos deberes*	*Porque tengo muchos deberes*	*Porque tengo muchos deberes*
2	*Porque los días de entrenamiento no me vienen bien*	*Porque los días de entrenamiento no me vienen bien*	*Porque los días de entrenamiento/actividades deportivas no me vienen bien*
3	*Porque mis amigos no practican actividad física*	*Porque mis amigos/as no practican actividad física*	*Porque mis amigos/as no practican deporte*
4	*Porque no tengo suficiente forma física*	*Porque no tengo buena forma física*	*Porque no estoy en buena forma física*
5	*Porque no me interesa la actividad física*	*Porque no me interesa la actividad física*	*Porque no me interesa el deporte*
6	*Porque me avergüenzo de mi cuerpo cuando practico actividad física*	*Porque me da vergüenza mi cuerpo cuando practico actividad física*	*Porque me da vergüenza mi cuerpo cuando practico deporte*
7	*Porque no disfruto con la actividad física*	*Porque no disfruto con la actividad física*	*Porque no disfruto con el deporte*
8	*Porque siento que mi aspecto físico es peor que los demás*	*Soy peor que los/las demás en el deporte*	*Soy peor que los/las demás en el deporte*
9	*Porque nadie me anima a hacer actividad física*	*Porque nadie me dice de hacer deporte*	*Porque nadie me dice de hacer deporte*
10	*Porque no hay actividades físicas que me gusten*	*Porque no hay deportes que me gusten*	*Porque no hay deportes que me gusten*
11	*Porque tengo miedo a hacer el ridículo*	*Porque tengo miedo a que se rían de mi*	*Porque tengo miedo a que se rían de mi*
12	*Porque considero que ya hago suficiente actividad física en las clases de educación física*	*Porque pienso que ya hago suficiente deporte en las clases de educación física*	*Porque pienso que ya hago suficiente deporte en las clases de educación física*

**Table 2 healthcare-13-02991-t002:** “Brief Scale of Perceived Barriers to Physical Activity for Children” goodness-of-fit indices.

Indices	Value
CMIN/DF	1.394
P (χ^2^)	0.040
RMSEA	0.054
CFI	0.976
TLI	0.966

CMIN/DF, minimum discrepancy per degree of freedom; P (χ^2^), chi-squared probability; RMSEA, root mean square error of approximation; CFI, comparative fit index; TLI, Tuker–Lewis index.

**Table 3 healthcare-13-02991-t003:** Reliability, test–retest, and systematic differences in the “Brief Scale of Perceived Barriers to Physical Activity for Children”.

Item	Test (n = 137)			Retest (n = 137)				Reliability Test					
	M	SD	Item-Total Correlation	M	SD	Item-Total Correlation	Cronbach’s α If Item Deleted	ICC (95% CI)	*p*-Value †	SEM	SEM%	MDC	MDC%
Item 1	1.61	0.93	0.537 **	1.61	0.99	0.568 **	0.930	0.869 (0.822 to 0.905)	0.862	0.35	21.6	0.96	59.8
Item 2	1.51	1.00	0.565 **	1.38	0.96	0.564 **	0.866	0.758 (0.675 to 0.822)	0.024	0.48	33.4	1.34	92.5
Item 3	1.24	0.66	0.488 **	1.27	0.66	0.555 **	0.928	0.865 (0.817 to 0.902)	0.317	0.24	19.3	0.67	53.6
Item 4	1.37	0.83	0.518 **	1.31	0.75	0.501 **	0.884	0.792 (0.720 to 0.847)	0.182	0.36	26.9	1.00	74.5
Item 5	1.33	0.89	0.552 **	1.29	0.94	0.517 **	0.952	0.909 (0.875 to 0.934)	0.275	0.28	21.1	0.77	58.4
Item 6	1.23	0.70	0.400 **	1.23	0.69	0.433 **	0.943	0.893 (0.853 to 0.922)	>0.999	0.23	18.5	0.63	51.2
Item 7	1.40	0.99	0.550 **	1.29	0.86	0.522 **	0.831	0.708 (0.613 to 0.782)	0.071	0.50	37.2	1.39	103.0
Item 8	1.53	1.04	0.608 **	1.55	1.10	0.570 **	0.924	0.860 (0.809 to 0.898)	0.763	0.40	26.0	1.11	72.1
Item 9	1.28	0.67	0.506 **	1.28	0.67	0.485 **	0.917	0.847 (0.792 to 0.888)	0.819	0.26	20.5	0.73	56.8
Item 10	1.42	1.00	0.518 **	1.45	1.06	0.575 **	0.981	0.962 (0.947 to 0.973)	0.132	0.20	14.0	0.56	38.8
Item 11	1.44	1.07	0.481 **	1.39	0.97	0.521 **	0.912	0.838 (0.781 to 0.882)	0.376	0.41	29.0	1.14	80.4
Item 12	1.41	1.02	0.515 **	1.42	1.03	0.586 **	0.968	0.938 (0.914 to 0.955)	0.637	0.26	18.0	0.71	50.0
Total score	16.77	7.16	N/A	16.48	6.94	N/A	0.979	0.979 (0.971 to 0.985)	0.098	1.02	6.1	2.83	17.0

Abbreviations: M, Mean; SD, standard deviation; 95% CI, confidence interval of 95%; ICC, intraclass correlation coefficient; SEM, standard error of measurement; %SEM, standard error of measurement as a percentage; MDC, minimum detectable change; N/A, not applicable; † Friedman test *p*-values. Cronbach’s α if item deleted indicates the internal consistency of the scale when each specific item is removed. Item-total correlation refers to the magnitude of association between each item with the total score. ** Significant correlation at *p* < 0.01.

## Data Availability

The dataset used and analyzed in this study is available from the corresponding author. The data are not publicly available due to privacy concerns.

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
