# Peer review of "Cross-Cultural Adaptation and Validation of the “Brief Scale of Perceived Barriers to Physical Activity for Children”: Analysis of Psychometric Properties"

_healthcare, 2025, doi:10.3390/healthcare13222991_

Round 1
Reviewer 1 Report
Comments and Suggestions for Authors
This study has the potential to make a significant contribution to the field by adapting and analyzing the validity and reliability of a short scale to identify perceived barriers to physical activity in Spanish school-aged children. The topic is timely and pedagogically valuable. However, methodological transparency and reporting need significant improvement before publication.

Author Response
This study has the potential to make a significant contribution to the field by adapting and analyzing the validity and reliability of a short scale to identify perceived barriers to physical activity in Spanish school-aged children. The topic is timely and pedagogically valuable. However, methodological transparency and reporting need significant improvement before publication.
Author’s response: We sincerely thank for your their positive and encouraging comments regarding the potential contribution and relevance of our study. Accordingly, we have reviewed and improved the manuscript according to your suggestions.
- Abstract: The abstract clearly states the purpose, method, results, and implications.
Introduction
- The introduction section should be restructured. The information in this section is generally too scattered and lacks a coherent flow. Extensive information should be provided on the positive effects of physical activity. Simply stating "its positive effects have been demonstrated in numerous studies" is insufficient. The second sentence includes information specific to Spain, followed by a mixed presentation of childhood and adolescence. After the justification for Spain is presented, comparisons can be made between childhood and adolescence. The introduction section is currently quite confusing.
Author’s response: We thank you for your observation. The introduction has been restructured to improve coherence and logical flow. We expanded the section on the positive effects of physical activity, presented the justification for Spain before comparing childhood and adolescence, and refined transitions to ensure greater clarity.
- The research hypothesis or research question should be clearly stated in the final paragraph.
Author’s response: Thank you for your suggestion. The research hypothesis has been included.
Materials and Methods
- The authors state that 17 children participated in the cognitive interview phase. However, the reason for choosing this number is unclear. Please justify the number of participants who participated in the cognitive interview phase (n = 17).
Author’s response: Thank you for your contribution. Following your suggestion, the pilot sample of children from the cognitive interviews has been justified in the text, in the materials and methods section.
- Although the study outlined the cultural adaptation of the scale, the forward-backward translation process from the original language and the steps taken to maintain semantic equivalence were not explained. A key component of cultural adaptation in scale adaptation studies is the forward-backward translation process. This process ensures that the original scale is adapted to the target language while maintaining semantic equivalence (e.g., Beaton et al., 2000; WHO Guidelines, 2018). Please indicate whether these steps were undertaken, and if so, how.
Author’s response: We appreciate your comment. The forward–backward translation process was not carried out because the original instrument was already developed in Spanish. Therefore, this study focused exclusively on the cultural adaptation of the scale to ensure its suitability for children aged 6 to 12 years, rather than on linguistic translation.
- The number of participants (n=137) is limited for CFA; justification for sample size (e.g., number of observations per factor) should be stated.
Author’s response: Thanks four your valuable appreciation. While it is acknowledged that larger sample sizes are often recommended for CFA, existing literature provides evidence that CFA can be effectively conducted with smaller sample sizes under certain conditions (Cruchinho et al., 2024; Van Prooijen & Van Der Kloot, 2001). In this regard, I can point to a study with a similar methodology that yielded good results from a sample of fewer than 137 participants (Mendoza-Muñoz et al., 2023). In our study, the CFA model is well-defined, with a clear factor structure and adequate indicator loadings, supporting the adequacy of the sample size. Additionally, no missing data were present in our dataset, eliminating concerns related to data imputation and preserving the effective sample size.
Cruchinho, P., López-Franco, M. D., Capelas, M. L., Almeida, S., Bennett, P. M., Miranda Da Silva, M., Teixeira, G., Nunes, E., Lucas, P., & Gaspar, F. (2024). Translation, Cross-Cultural Adaptation, and Validation of Measurement Instruments: A Practical Guideline for Novice Researchers. Journal of Multidisciplinary Healthcare, Volume 17, 2701-2728. https://doi.org/10.2147/JMDH.S419714
Mendoza-Muñoz, M., Carlos-Vivas, J., Castillo-Paredes, A., Parraca, J. A., Raimundo, A., Alegrete, J., Pastor-Cisneros, R., & Gomez-Galan, R. (2023). Portuguese translation and validation of the questionnaires from the Canadian Physical Literacy Assessment-2: A pilot study. Frontiers in Psychology, 14, 1244566. https://doi.org/10.3389/fpsyg.2023.1244566
Van Prooijen, J.-W., & Van Der Kloot, W. A. (2001). Confirmatory Analysis of Exploratively Obtained Factor Structures. Educational and Psychological Measurement, 61(5), 777-792. https://doi.org/10.1177/00131640121971518
- Clearly state in the Limitations section that convenience sampling was used. While this method is suitable for validity studies, it is essential to note that it limits the generalizability of the results.
Author’s response: We value your comment. We have revised the Limitations section to explicitly indicate that convenience sampling was used. We acknowledge that it may limit the generalizability of the findings.
- Showing all the items in Table 1 is nice, but it's too detailed. The final version of the scale is already included in the supplementary file. Therefore, the table should either be moved to the supplementary file or summarized with just a few sample items.
Author’s response: We thank your suggestion. We prefer to keep all items in Table 1 in the main manuscript to facilitate direct access to the full scale for the reader. However, if the reviewer considers it strictly necessary, we are willing to move the table to the supplementary material.
Results
- CFA results are repeated in the text; unnecessary repetition could be reduced.
Author’s response: We value your comment. Following your suggestion, some specific values have been removed to avoid repetition. However, to ensure that no relevant information is inadvertently removed, we kindly ask you to specify which parts of the CFA results they consider repetitive. This will allow us to reduce unnecessary repetition while retaining all essential content.
- Criterion validity or lack of external validity should be emphasized in the limitations section.
Author’s response: We thank your observation. We have revised the Limitations section to explicitly acknowledge the absence of criterion validity and the resulting limitation in terms of external validity.
Conclusion
- The conclusion is quite long; it serves as a repetition of the discussion. A shorter, more focused paragraph is recommended.
Author’s response: Following your suggestion, we have shortened and focused the Conclusion section to avoid repetition of the Discussion, while retaining the key findings and implications of the study.
Reviewer 2 Report
Comments and Suggestions for Authors
- Abstract
The abstract section has several requirement revision.
Avoiding the mixing of methods and indicators, “Analyses included confirmatory factor analysis (CFA), Cronbach’s alpha (α), and intraclass correlation coefficient (ICC). ”
- The culturally adapted tool shouldperform EFA before conducting CFA. Additionally, it is imperative to specify the sample size utilized for the CFA.
- The internal consistency and test-retest reliability results require precise explanation.
- The conclusion has been overly expanded. It is recommended to focus on the current research subject.
- For the keywords, it is recommended to choose"cultural adaptation" and "psychometric properties" as they can better summarize the research.
- Background
The introduction would be strengthened by a more detailed explanation for the choice of Brief Scale of Perceived Barriers to Physical Activity for Children. I recommend that the authors include a brief narrative review of the existing tools used to assess physical activity barriers in children. This review should explicitly state which tools have already been validated or not validated in Spanish and, more importantly, identify the gap that this study aims to fill by adapting and validating this particular instrument. This will make a much clearer case for the study's significance and why you selected the "Brief Scale of Perceived Barriers to Physical Activity for Children" for cultural adaptation and validation in this study.
- Method
- The authors state that the original scale was modified in "vocabulary, syntax, and content." When such substantial adaptations are made, the resulting instrument is essentially new, necessitating an exploratory (EFA) approach to identify the factor structure in one cohort, followed by confirmatory (CFA) testing in a separate sample. With the current limited sample, the author should explain that why it is reasonable to skip EFA and directly proceed with CFA.
- The description of the pilot survey is insufficient. To establish that the tool is acceptable and comprehensible to children across different developmental stages within the school-age range, it is essential to detail how the 17 participants were recruited and who they were. Please provide a detailed paragraph summarizing their key characteristics.
- Results
- Figure 1 indicates that the authors introduced theoretically justifiable error covariances between Item 1 and 4, and between Item 2 and 3. To ensure the transparency and scientific rigor of the CFA process, it is essential for the authors to present the model fit indices (e.g., χ², CFI, TLI, RMSEA, SRMR) for the initial model (without the error covariances). Furthermore, in the results or discussion section, they should provide a clear theoretical or substantive explanation for the covariation between these specific item pairs.
- The information currently presented in Table 1 (CFI, TLI, RMSEA) is sufficiently straightforward to be described concisely in the text.
- For table 2, there are some confusing issues. Firstly, Cronbach's alpha is a measure of internal consistency for a scale or subscale, not for individual items. The authors should correct the manuscript by reporting alpha values only for the total scale and its dimensions. Secondly, as the authors demonstrated in lines 106-108, "Additionally, a test-retest reliability study was carried out with a subsample of children who completed the questionnaire twice, with a two-week interval, to assess the temporal stability of the responses." The authors mention using a subset of the sample for test-retest reliability measurement, but the results presented in Table 2 show that 137 participants completed both the initial and retest assessments, leading to a contradiction in the main findings of Table 2.
- There are currently two table1 in the manuscript. There are still errors in the manuscript due to carelessness. It is recommended to review the entire document.
Discussion
It is recommended that the author address the inconsistencies between the methodology and results sections, as this is essential to ensuring the scientific validity of the article. Corresponding revisions should then be made to the discussion section.
Comments on the Quality of English LanguageThe English could be improved to more clearly express the research.
Author Response
We sincerely thank for your their positive and encouraging comments regarding the potential contribution and relevance of our study. Accordingly, we have reviewed and improved the manuscript according to your suggestions.
- Abstract
The abstract section has several requirement revision.
Avoiding the mixing of methods and indicators, “Analyses included confirmatory factor analysis (CFA), Cronbach’s alpha (α), and intraclass correlation coefficient (ICC). ”
- The culturally adapted tool shouldperform EFA before conducting CFA. Additionally, it is imperative to specify the sample size utilized for the CFA.
- The internal consistency and test-retest reliability results require precise explanation.
- The conclusion has been overly expanded. It is recommended to focus on the current research subject.
- For the keywords, it is recommended to choose"cultural adaptation" and "psychometric properties" as they can better summarize the research.
Author’s response: We sincerely thank for your constructive feedback and valuable suggestions. Regarding the last two points, the Conclusion subsection has been shortened to focus exclusively on the current research subject, and the keywords “cultural adaptation” and “psychometric properties” have been added, as suggested. We greatly appreciate this recommendation.
With respect to the exploratory analysis, an Exploratory Factor Analysis (EFA) was not performed because the original instrument already presented a well-established factorial structure with clearly defined dimensions. Therefore, we directly conducted a Confirmatory Factor Analysis (CFA) to verify the fit of this existing model in the Spanish context, following standard procedures in cultural adaptation studies where the original construct has been previously validated.
We believe that the abstract now includes the essential and precise information on reliability and validity results. Further expansion would exceed the word limit established by the journal for the abstract.
Once again, we appreciate your helpful comments, which have substantially improved the clarity and focus of our manuscript.
- Background
The introduction would be strengthened by a more detailed explanation for the choice of Brief Scale of Perceived Barriers to Physical Activity for Children. I recommend that the authors include a brief narrative review of the existing tools used to assess physical activity barriers in children. This review should explicitly state which tools have already been validated or not validated in Spanish and, more importantly, identify the gap that this study aims to fill by adapting and validating this particular instrument. This will make a much clearer case for the study's significance and why you selected the "Brief Scale of Perceived Barriers to Physical Activity for Children" for cultural adaptation and validation in this study.
Author’s response: We thank for your insightful comment and fully agree that a clearer justification for the selection of the Brief Scale of Perceived Barriers to Physical Activity for Children strengthens the rationale of the study.
To address this, the revised version now includes a short narrative review of existing instruments assessing perceived barriers to physical activity in children. To the best of our knowledge, there are no validated tools in Spanish specifically designed to measure perceived barriers to physical activity in children.
The instrument chosen for this study, the Brief Scale of Perceived Barriers to Physical Activity for Children, was originally developed for adolescents. Therefore, a cultural and developmental adaptation was required to make it appropriate for the child population. One of its main advantages is that it is already available in Spanish, so a full translation process was not necessary, ensuring linguistic consistency and cultural familiarity. Additionally, this scale includes a small number of items, making it concise and practical to administer—an important feature since it will be integrated into a broader Spanish Physical Literacy Assessment model (SPLA-C).
Following your suggestion, we have also reviewed other instruments reported in the literature. To our knowledge, only one other questionnaire specifically aimed at assessing facilitators and barriers to physical activity in children has been identified (Arlinghaus et al., Medicine & Science in Sports & Exercise, 2021; link). However, this tool has not been validated in Spanish and contains a larger number of items and subscales, which we considered less suitable for inclusion within a multidimensional assessment model such as the SPLA-C.
Therefore, our choice of the Brief Scale of Perceived Barriers to Physical Activity for Children was guided by both practical and methodological considerations—its brevity, prior use in Spanish-speaking contexts, and the need for an age-appropriate, easily administered tool that aligns with the structure of the SPLA-C.
- Method
- The authors state that the original scale was modified in "vocabulary, syntax, and content." When such substantial adaptations are made, the resulting instrument is essentially new, necessitating an exploratory (EFA) approach to identify the factor structure in one cohort, followed by confirmatory (CFA) testing in a separate sample. With the current limited sample, the author should explain that why it is reasonable to skip EFA and directly proceed with CFA.
Author’s response: We appreciate your comment. In this case, however, we would like to clarify that the adaptations made to the original scale were minimal and not substantial. The adjustments were limited to minor vocabulary and syntactic changes to ensure developmental and linguistic appropriateness for younger children, without altering the conceptual meaning or structure of the items. Therefore, the content validity and theoretical framework of the original instrument remained intact.
Given that the Brief Scale of Perceived Barriers to Physical Activity for Children already has a validated and well-established factorial structure, it was deemed methodologically appropriate to proceed directly with a Confirmatory Factor Analysis (CFA) to test whether this existing model fit the Spanish child population. Conducting an EFA was not necessary, as the goal was to verify the cross-cultural and developmental applicability of an existing, theoretically supported model rather than to explore a new factorial structure.
We apologize for any confusion caused by the initial wording in the manuscript. The text has now been clarified to specify that the modifications were not substantial and did not justify an exploratory approach.
- The description of the pilot survey is insufficient. To establish that the tool is acceptable and comprehensible to children across different developmental stages within the school-age range, it is essential to detail how the 17 participants were recruited and who they were. Please provide a detailed paragraph summarizing their key characteristics.
Author’s response: Thank you for your contribution. Following your suggestion, we have added a detailed paragraph summarising the key characteristics that you mentioned, in the material and methods section.
- Results
- Figure 1 indicates that the authors introduced theoretically justifiable error covariances between Item 1 and 4, and between Item 2 and 3. To ensure the transparency and scientific rigor of the CFA process, it is essential for the authors to present the model fit indices (e.g., χ², CFI, TLI, RMSEA, SRMR) for the initial model (without the error covariances). Furthermore, in the results or discussion section, they should provide a clear theoretical or substantive explanation for the covariation between these specific item pairs.
Author’s response: We sincerely thank the reviewer for this insightful and constructive comment.
To enhance the transparency of the confirmatory factor analysis (CFA) process, we have prepared a table (shown below) presenting the model fit indices (χ², CFI, TLI, RMSEA, and SRMR) for both the initial model (without correlated errors).
If the reviewer considers that this information should be incorporated into the manuscript, we would be pleased to include it — either as a separate table or integrated with the existing psychometric results in Table 1 (Table 2 now). We would greatly appreciate any guidance or reference the reviewer could provide on how to present these data jointly, to ensure that the information is clear and not redundant or confusing for readers.
|
Indices |
Value |
|
CMIN/DF |
1.998 |
|
P valor |
<0.001 |
|
RMSEA |
.086 |
|
CFI |
.937 |
|
TLI |
.914 |
|
SRMR |
.0534 |
SRMR value with adjustments (=.0445)
- The information currently presented in Table 1 (CFI, TLI, RMSEA) is sufficiently straightforward to be described concisely in the text.
Author’s response: Thank you for your contribution. Specific values have been removed to avoid repetition. However, if you believe that further information should be removed, please let us know.
- For table 2, there are some confusing issues. Firstly, Cronbach's alpha is a measure of internal consistency for a scale or subscale, not for individual items. The authors should correct the manuscript by reporting alpha values only for the total scale and its dimensions. Secondly, as the authors demonstrated in lines 106-108, "Additionally, a test-retest reliability study was carried out with a subsample of children who completed the questionnaire twice, with a two-week interval, to assess the temporal stability of the responses." The authors mention using a subset of the sample for test-retest reliability measurement, but the results presented in Table 2 show that 137 participants completed both the initial and retest assessments, leading to a contradiction in the main findings of Table 2.
Author’s response: We appreciate your comment. However, we would like to clarify that item-level alpha values were also computed and presented not as independent reliability coefficients, but as diagnostic indicators to assess the contribution of each item to the internal consistency of its subscale. This procedure is commonly applied even in the context of CfA, as it helps to identify potentially weak or redundant items that may require refinement or removal while maintaining the theoretically predefined factor structure.
Regarding the test–retest reliability results, we confirm that the subsample consisted of 137 participants, which represents a subset of the total sample. The text in the Methods section has been clarified accordingly to ensure consistency with Table 2.
- There are currently two table 1 in the manuscript. There are still errors in the manuscript due to carelessness. It is recommended to review the entire document.
Author’s response: Amended.
Discussion
It is recommended that the author address the inconsistencies between the methodology and results sections, as this is essential to ensuring the scientific validity of the article. Corresponding revisions should then be made to the discussion section.
Author’s response: We value your comment. We have carefully reviewed the manuscript in light of the previous comments, and all methodological clarifications have been incorporated. However, we would greatly appreciate if the reviewer could kindly specify which particular inconsistencies are being referred to, or if they have already been resolved with the recent modifications. This clarification will help us ensure that all remaining issues are fully addressed and that the final version meets the expected scientific standards.
Reviewer 3 Report
Comments and Suggestions for Authors
The article is very interesting in the area of health literacy among adolescents, a group that needs tools to be developed in this area.
In the methodology, the authors mention that they performed a psychometric analysis of the scale, but only refer to confirmatory analysis. However, exploratory analysis should be performed before confirmatory analysis. Why did they not perform exploratory factor analysis? Do they immediately consider that the structure of the scale in their sample is the same as that of the original scale? The authors should clarify this.
The methodology also does not mention the calculation of the sample size; this issue is essential for psychometric analysis. What sample size do you consider acceptable for our study? Did you make any formal calculations?
The results should include a table with the general characteristics of the sample for better interpretation of the study.
Author Response
The article is very interesting in the area of health literacy among adolescents, a group that needs tools to be developed in this area.
Author’s response: We sincerely thank for your their positive and encouraging comments regarding the potential contribution and relevance of our study
In the methodology, the authors mention that they performed a psychometric analysis of the scale, but only refer to confirmatory analysis. However, exploratory analysis should be performed before confirmatory analysis. Why did they not perform exploratory factor analysis? Do they immediately consider that the structure of the scale in their sample is the same as that of the original scale? The authors should clarify this.
Author’s response: We value your comment. Given that the Brief Scale of Perceived Barriers to Physical Activity for Children already has a validated and well-established factorial structure, it was deemed methodologically appropriate to proceed directly with a Confirmatory Factor Analysis (CFA) to test whether this existing model fit the Spanish child population. Conducting an EFA was not necessary, as the goal was to verify the cross-cultural and developmental applicability of an existing, theoretically supported model rather than to explore a new factorial structure.
The methodology also does not mention the calculation of the sample size; this issue is essential for psychometric analysis. What sample size do you consider acceptable for our study? Did you make any formal calculations?
Author’s response: Thank you very much for your comment. The sample size for this study was determined based on standard psychometric recommendations. Specifically, it is commonly suggested to include 5–10 participants per item for factor analysis (Hair et al., 2010). Considering that our instrument contains 12 items, the minimum recommended sample would be 60–120 participants.
For confirmatory factor analysis (CFA), it is generally advised to use a larger sample (100–200 participants) to ensure stability of model fit indices and reliable estimation of factor loadings. Our total sample exceeded this recommendation (n = 137), providing a sufficiently powered analysis. In this regard, I can point to a study with a similar methodology that yielded good results from a sample of fewer than 137 participants (Mendoza-Muñoz et al., 2023).
In our study, 137 participants completed both administrations, providing robust and precise reliability estimates.
No formal statistical sample size calculation was performed a priori, as the study followed conventional psychometric guidelines based on item-to-participant ratios and CFA requirements.
Hair, J. F., Black, W. C., Babin, B. J., & Anderson, R. E. (2010). Multivariate Data Analysis (7th ed.). Pearson.
Mendoza-Muñoz, M., Carlos-Vivas, J., Castillo-Paredes, A., Parraca, J. A., Raimundo, A., Alegrete, J., Pastor-Cisneros, R., & Gomez-Galan, R. (2023). Portuguese translation and validation of the questionnaires from the Canadian Physical Literacy Assessment-2: A pilot study. Frontiers in Psychology, 14, 1244566. https://doi.org/10.3389/fpsyg.2023.1244566
The results should include a table with the general characteristics of the sample for better interpretation of the study.
Author’s response: We sincerely you for this valuable suggestion.We would like to clarify that all relevant information regarding the participants is already provided in the text, including gender distribution, age range, and the total number of participants. This information is straightforward, concise, and clearly presented, allowing readers to understand the sample without the need for a separate table.
However, if there are additional characteristics of the sample that you believes should be included, or if they consider that presenting the existing information in a tabular format would enhance clarity, we would be pleased to make these changes. At present, we believe that including a table might be repetitive, given the brevity and simplicity of the information, and that presenting it in the text is sufficient for interpretation.
Round 2
Reviewer 2 Report
Comments and Suggestions for Authors
The authors are commended for their diligent revisions, which have significantly enhanced the methodological completeness and transparency of the manuscript. These changes have strengthened the robustness and clarity of the presented findings. But there are still there major points that require special attention.
1.Regarding the revisions for the previous version, the author failed to make the corrections correctly in Abstract section.
To avoid the confusion of methods and indicators, "The analyses included confirmatory factor analysis (CFA), Cronbach's alpha (α), and intraclass correlation coefficient (ICC)." I recommend making the following modifications, for example, “Several analyses were conducted, including confirmatory factor analysis (CFA) to assess the factor structure, along with reliability metrics: Cronbach’s alpha (α) for internal consistency and the intraclass correlation coefficient (ICC) for test-retest reliability.”
2.Concerning table 2.
In summary, while the underlying analysis is appropriate, the "item-level alpha" is scientifically incorrect. In the methodological literature on scale development and validation, the term "Cronbach's alpha if item deleted" is the universally recognized and accepted terminology for this diagnostic index. So, I speculate whether the meaning you intend to convey is "Cronbach's alpha if an item is deleted"?The term you have used may be ambiguous and can be misleading to readers. We kindly request that you implement this change in the text and table 2.
3.Concerning Fig 1, the manuscript reports the use of "theoretically justified error covariances" between specific item pairs (Item 1 & 4 and Item 2 & 3) to improve model fit in the confirmatory factor analysis. However, the justification for selecting these particular pairs is not provided in the methods, results, or discussion sections.
Simply stating that the covariances are "theoretically justified" is insufficient without explaining the specific theory or reason. Therefore, the authors must provide a clear and explicit explanation for each specified error covariance: For the pair Items 1 & 4 and Items 2 & 3: What is the specific theoretical, conceptual, or methodological reason that justifies the shared variance between these specific items, and not others?
Author Response
We thank the reviewer for the careful reading and constructive suggestions. We have addressed each point and revised the manuscript accordingly. Changes are highlighted in the revised manuscript file and described below.
1.Regarding the revisions for the previous version, the author failed to make the corrections correctly in Abstract section. To avoid the confusion of methods and indicators, "The analyses included confirmatory factor analysis (CFA), Cronbach's alpha (α), and intraclass correlation coefficient (ICC)." I recommend making the following modifications, for example, “Several analyses were conducted, including confirmatory factor analysis (CFA) to assess the factor structure, along with reliability metrics: Cronbach’s alpha (α) for internal consistency and the intraclass correlation coefficient (ICC) for test-retest reliability.”
Author's response: Thank you for your valuable comment. We have corrected the Abstract to more clearly describe the analyses and the role of each indicator. We replaced the original sentence with the following wording suggested by the reviewer
2. Concerning table 2. In summary, while the underlying analysis is appropriate, the "item-level alpha" is scientifically incorrect. In the methodological literature on scale development and validation, the term "Cronbach's alpha if item deleted" is the universally recognized and accepted terminology for this diagnostic index. So, I speculate whether the meaning you intend to convey is "Cronbach's alpha if an item is deleted"?The term you have used may be ambiguous and can be misleading to readers. We kindly request that you implement this change in the text and table 2.
Author's response: We appreciate your comment. We replaced the ambiguous term throughout the manuscript and in Table 2.
3. Concerning Fig 1, the manuscript reports the use of "theoretically justified error covariances" between specific item pairs (Item 1 & 4 and Item 2 & 3) to improve model fit in the confirmatory factor analysis. However, the justification for selecting these particular pairs is not provided in the methods, results, or discussion sections. Simply stating that the covariances are "theoretically justified" is insufficient without explaining the specific theory or reason. Therefore, the authors must provide a clear and explicit explanation for each specified error covariance: For the pair Items 1 & 4 and Items 2 & 3: What is the specific theoretical, conceptual, or methodological reason that justifies the shared variance between these specific items, and not others?
Author's response: Thank you for pointing out this important aspect. We have verified that the pairs cited by the reviewer (1 and 4, and 2 and 3) refer to items 6 and 8, and items 11 and 4 (as shown in Figure 1 of the revised manuscript). the correlated residuals are actually found between the following pairs of items within the ‘Self-concept’ factor:
Item 8 (‘I am worse than others at sports’) and item 6 (‘Because I am ashamed of my body when I play sports’).
Item 11 (‘Because I am afraid they will laugh at me’) and item 4 (‘Because I am not in good physical shape’).
We have corrected the manuscript and added explicit theoretical justifications.